# Faithfulness Hallucination Detection in Healthcare AI

Prathiksha Rumale V[1,*], Simran Tiwari[2,*], Tejas G Naik[1,*], Sahil Gupta[1,*], Dung N Thai[2,*], Wenlong Zhao[1,*], Sunjae Kwon[1], Victor Ardulov[2], Karim Tarabishy[2], Andrew McCallum[1], Wael Salloum[2]

[1]University of Massachusetts Amherst, [2]Mendel AI, [*]Equal contributions.
{prumalevishw,tnaik,sahgupta,wenlongzhao}@umass.edu,{simran.t,dung.t}@mendel.ai

## Abstract

Faithfulness hallucinations, where AI-generated contents diverge from input contexts, pose significant risks in high-stakes environments like healthcare. In clinical settings, the reliability of systems is crucial, as any deviations can lead to misdiagnoses and inappropriate treatments. The burden of summarizing lengthy electronic health records (EHRs) is substantial for clinicians, who face the challenge of extracting key information from extensive notes. Leveraging large language models (LLMs) to summarize these documents can aid clinicians by providing concise and accurate summaries. This study investigates faithfulness hallucinations in medical record summaries generated by LLMs such as GPT-4o and Llama-3. Our detection framework, developed in collaboration with clinicians and supported by a web-based annotation tool, categorizes 5 types of medical event hallucinations. A pilot study involving 100 summaries of medical notes reveals the presence of our categorized hallucinations by recent closed-source and open-source LLMs. The findings highlight the necessity for robust hallucination detection methods to ensure reliability of AI applications in healthcare, ultimately enhancing clinical workflows and improving patient care.

## CCS Concepts

• **Computing methodologies → Natural language generation**; • **Applied computing → Health informatics**; • **General and reference → Cross-computing tools and techniques**; *Computing standards, RFCs and guidelines*; • **Social and professional topics → Personal health records**; • **Human-centered computing →** *Human computer interaction (HCI)*.

## Keywords

AI in Healthcare, Hallucination Detection, Generative Language Models, Natural Language Processing

**ACM Reference Format:**
Prathiksha Rumale V[1,*], Simran Tiwari[2,*], Tejas G Naik[1,*], Sahil Gupta[1,*], Dung N Thai[2,*], Wenlong Zhao[1,*], Sunjae Kwon[1], Victor Ardulov[2], Karim Tarabishy[2], Andrew McCallum[1], Wael Salloum[2]. 2024. Faithfulness Hallucination Detection in Healthcare AI. In *Proceedings of KDD 2024 Workshop - Artificial Intelligence and Data Science for Healthcare: Bridging Data-Centric AI and People-Centric Healthcare (KDD-AIDSH 2024)*. ACM, New York, NY, USA, 8 pages. https://doi.org/XXXXXXX.XXXXXXX

## 1 Introduction

The rapid advances in large language models (LLMs) such as Llama-3 [16] and GPT-4o [1] have captivated the world with their capabilities. Despite being trained as general purpose language models, they can often generate meaningful texts about medical contents, pass medical exams [8, 10, 13], and perform remarkably well on multiple healthcare and medical tasks [5, 11]. These LLMs can generate summaries, abstract patient information, and answer questions about medical records, demonstrating their potential to assist medical professionals on mundane and time-consuming problems. However, there is an unsolved limitation that these models may hallucinate, producing well-written but nonsensical and unfaithful content. While such hallucinations might be inconsequential in casual conversations, they pose significant risks in healthcare applications where soundness and trustworthiness are crucial. Inaccuracies in high-stakes healthcare settings can lead to severe consequences, including misdiagnoses and inappropriate treatments.

A critical challenge in achieving reliable AI in healthcare is addressing model hallucinations. Much existing research focuses on *factual hallucination*, where LLM generated information is inconsistent with verifiable real-world knowledge. Factually plausible descriptions, however, may still be harmful and misleading, if they are not grounded in the input context. *Faithfulness hallucination* refers to the phenomenon that models' generated content diverges from the instruction or context provided. This is most typically observed when LLMs produce summaries containing information that contradicts with or is not evident in the original document. Since medical records are highly contextualized and personalized, faithfulness hallucination is of paramount importance to ensuring the safe applications of LLMs in healthcare.

In this work, we propose the task of detecting faithfulness hallucinations in medical record summaries. We define 5 medical event hallucination types, incorrect reasoning, and chronological inconsistency, according to which a system should identify unfaithfulness between the original medical record and its summary. To facilitate hallucination annotation, we collaborate with clinicians to carefully craft data annotation guidelines and develop a web-based user interface. We have completed a pilot study using the proposed annotation tool to collect human expert annotations on 50 medical notes and their summaries generated by GPT-4o and Llama-3. Our findings indicate that hallucinations occur within the summaries from both closed-source and open-source LLMs, further confirming the need for hallucination detection in the medical domain.

Annotating hallucinations in medical record summaries by human experts is expensive, ushering in the need for an automatic hallucination detection system. In our pilot study, it took 92 minutes on average for a well-trained clinician to label a summary. At the rate of $36 per hour, annotating 100 summaries in our pilot study cost $5508 and 153 working hours in total. To this end, we establish

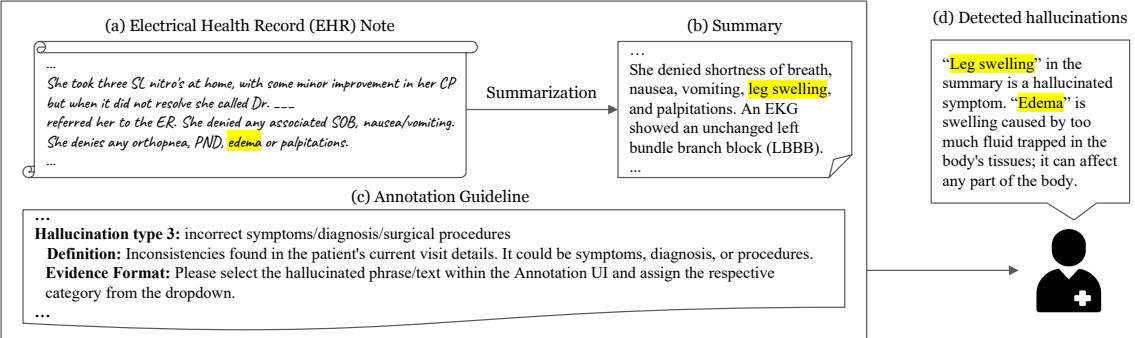

**Figure 1: Overview of the faithfulness hallucination detection task. Given a medical note (a) and its summary (b) by a large language model, a human expert or an automated system identifies hallucinations (d) in the summary according to our proposed hallucination annotation guideline (c). We then analyze the hallucination patterns of different summarization models.**

an extraction-based system and an LLM-based system as baselines for the task. We evaluate the two baselines on the 100 annotated summaries and analyze their strengths and limitations.

Our contributions are as follows:
(1) We propose the task of faithfulness hallucination detection in medical record summaries and categorize hallucinations types to enable a fine-grained evaluation of facts essential to various medical domains and patient care.
(2) We craft data annotation guidelines in collaboration with clinicians and develop a web-based user interface to collect hallucination annotations that are clinically relevant and accurate.
(3) We use the proposed annotation tool to collect clinician annotations on 100 medical notes and their summaries generated by GPT-4o and Llama-3, confirming the presence of various types of hallucinations in both closed-source and open-source LLMs.
(4) We explore the automation of hallucination detection by an extraction-based system and an LLM-based system on our labeled dataset to overcome the cost and time requirements of human hallucination annotation.

## 2 Related Work

**Faithfulness hallucination.** [11] explores trust challenges in LLMs by focusing on factual accuracy and faithfulness, emphasizing alignment between model outputs and instructions, context, and logic. The paper delves into the root causes of hallucinations in LLMs, including flawed data, biased training, and limitations during inference. It also introduces the concept of "faithfulness hallucinations," where model outputs deviate from inputs. To assess the prevalence of hallucination, benchmarks like TruthfulQA [12], MedHALT [15] and REALTIMEQA [9] have been established and these efforts aim to improve the reliability of LLMs in real-world applications. Our work builds upon this foundation and study the detection of faithfulness hallucinations in healthcare AI.

**Faithfulness hallucination in medical summaries.** A pioneering study [4] investigates faithfulness hallucination in medical text summaries generated by the LED model [7] and argues that model hallucinations have critical impact on patient care. The authors manually extract standard concepts and actions and compound concepts from summaries and let clinical practitioners classify them as "Incorrect", "Missing details", or "Not in Notes". Realizing the specialty of the clinical domain, we categorize a set of 5

hallucination types to be detected, development annotation guidelines and tools, and evaluate hallucinations in summaries from the state-of-the-art LLMs.

**Evaluating medical tasks.** [6] features a systematic review of 519 studies published between Jan. 1st, 2022, and Feb. 19th, 2024, sourced from PubMed and Web of Science, and highlights the lack of standardized task definitions and dimensions of evaluation in the field. These studies were categorized based on evaluation data type, healthcare task, NLP/NLU task, evaluation dimension, and medical specialty. The findings emphasize the necessity of using real patient care data to assess LLMs' effectiveness in administrative tasks. Moreover, the evaluation criteria must extend beyond accuracy to encompass fairness, bias, toxicity, robustness, and deployment considerations across various medical specialties. [14] highlights the limitations of automatic metrics for NLG tasks in the clinical domain. Our work employs human experts to assess hallucinations in LLM-produced summaries. We contribute a guideline for human experts to detect and categorize hallucinations, allowing for more nuanced analysis compared to simple presence-absence judgments.

**Evaluating medical summaries.** [3] establishes a framework for studying "hospital course summarization." The work focuses on the task of automatically generating concise summaries of patient hospital stays using discharge summaries as a reference. The creation of the CLINSUM dataset [2], a large dataset of patient records with corresponding summaries, helps in the development and evaluation of summarization models. One of their mentioned future works is the development of evaluation methods that assess not only factual accuracy and coherence but also check for clinical relevance. We study the detection of faithfulness hallucinations detect hallucinations in the medical record summaries.

## 3 Faithfulness Hallucination Detection in Health AI

### 3.1 Task Definition

**Faithfulness hallucination in clinical note summaries** refers to the phenomena where summarization models generate content that is incorrect or too general according to information in the source clinical notes. These hallucinations can lead to serious consequences in patient care as they undermine the reliability and accuracy of clinical documentation. **Incorrectness** describes any

content in the summary that does not appear in the clinical note or is twisted. This includes generalized information in the medical notes that becomes incorrectly specific in the summary. **Specific ⇒ general** refers to any content in the summary that are correct but too general compared to the clinical note, such as, medical conditions that are described as hypernyms.

Given a medical record $d$, its corresponding summary $s$ (possibly generated by a summarization model), and a predefined set of hallucination categories $\mathcal{H}$, the objective of the **faithfulness hallucination detection task** is to detect hallucination occurrences $y$, providing a piece of textual evidence and predicting its hallucination category $h \in \mathcal{H}$. The evidence is in the form of a text span in the summary when possible. If the text span is unavailable or not comprehensible, the evidence is in the form of free-text explanations.

## 3.2 Hallucination Categories

Defining hallucination categories that fit all medical tasks is challenging. The variables of interest can change from one study to another, based on the available information in the medical records and the interests of the summary readers. In this work, we focus on discharge summaries. We collaborate with the clinical team to define a set of hallucination categories based on the structure of the medical notes.

**Medical event inconsistency.** We consider 5 types of hallucinations about different medical events. (1) Patient Information: Hallucinated demographic details and non-medical information about the patient's background including name, age, gender, ethnicity, race, and address. (2) Patient History: Hallucinated information regarding the history of present illness. Many instances of hallucination include incorrectly stated illness or previously unseen health conditions. (3) Symptoms/Diagnosis/Surgical Procedures: Inconsistent symptoms, diagnosis, or procedures found in the patient's current visit details. (4) Medicine Related Instructions: Any disparities or discrepancies noted between the medication instructions documented in the summary and those found in the medical note. (5) Follow-up: Missing information regarding "follow-up" care or instructions provided to the patient. This includes appoint rescheduling and continued monitoring. Examples sections that may have follow-up information include *discharge diagnosis*, *discharge condition*, and *discharge instructions*.

**Chronological Inconsistency.** The order of medical events is not consistent with the sequence documented in the EHR. For example, an event that supposedly happened after another event is described as occurring before it.

**Incorrect Reasoning.** Summary states correct information but the associated reasoning given for it does not make sense or is incorrect.

## 4 Data Annotation

## 4.1 Guidelines

We proposed our initial guidelines based on the structure of the medical notes and identified potential hallucination types in each part. The guidelines included positive and negative examples for each hallucination type. We recruited a team of 15 clinicians. Each clinician had at least 5 years of experience abstracting medical records

and building clinical ontologies. The team provided feedback on the chosen examples and categorized hallucination types. Following trail runs and subsequent revisions, the guidelines grouped the hallucinations into the presented categories based on their impact.

## 4.2 Web-based UI

We developed a web-based data annotation user interface (UI) that allows annotators to highlight hallucinated spans from summaries and label their categories. The tool also allows annotators to provide free-text explanations for the detected hallucinations. Key features of the data annotation tool are highlighted in Figure 2. The tool lists extracted spans for annotators to review and amend if necessary. Annotations are automatically uploaded once submitted. The tool also supports loading previously annotated results for consensus and quality control.

## 4.3 Summary Collection

For our pilot study, we utilized a dataset consisting of 50 medical notes and their corresponding summaries generated by GPT-4o and Llama-3. These medical notes were randomly sampled from the MIMIC-IV (Medical Information Mart for Intensive Care IV) database, specifically focusing on discharge summaries. The selected notes provided a diverse set of patient records, ensuring a representative sample of typical discharge summaries. Each note contained comprehensive details about the patient's medical history, current visit information, symptoms, diagnosis, treatment plans, medication instructions, and follow-up instructions. GPT-4o and Llama-3 were prompted to summarize the medical notes with specific instructions to provide any available information regarding the variables of interest. To promote brevity, the models were additionally prompted to generate the summaries of no more than $n = 500$ words. Explicitly the prompt provided was:

> **Summarization Prompt**
>
> Summarize the provided clinical note with at most {n} words. Ensure to capture the following essential information, when they exist: the specific cancer type, its morphology, cancer stage, progression, TNM staging, prescribed medications, diagnostic tests conducted, surgical interventions performed, and the patient's response to treatment. {clinical note}

## 4.4 Hallucination Annotation

Following a training session to familiarize themselves with the annotation guidelines and tool, each clinician was instructed to identify and annotate inconsistencies between the medical notes and their summaries, labeling them according to our hallucination categories. Each medical event inconsistency is also labeled as either *incorrect* or *specific=>general*. Clinicians began the annotation process by reading medical records using a search tool for abstraction and extraction of medical events, followed by reading summaries to label hallucinations as per guidelines. They verified suspicious information by cross-referencing the records and summaries. The average annotation time per note was 91.5 minutes. A quality control (QC) member reviewed and corrected annotations, calculating

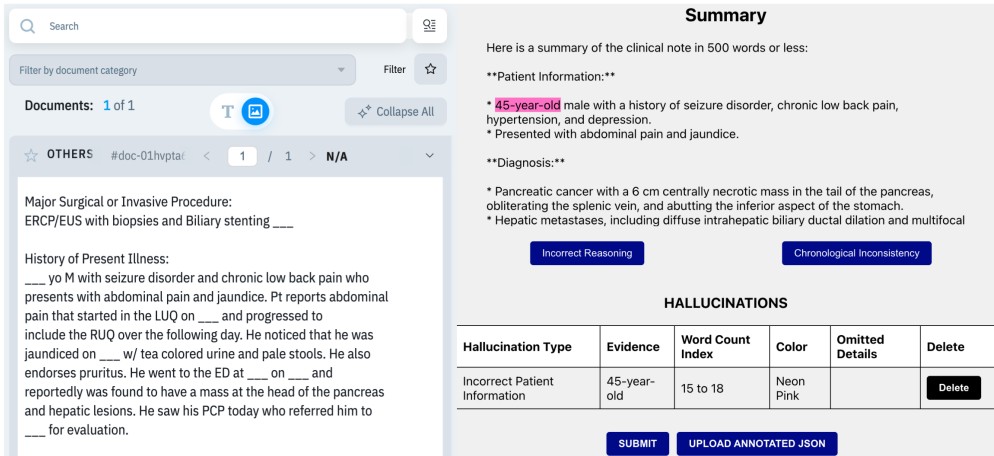

**Figure 2: The annotator tool assists annotators in identifying and selecting text spans that contain hallucinated content.**

a quality score from a random 5% sample per annotator. A detailed step by step procedure of the annotation process is described in Appendix A.

## 5  Hallucination Pattern Analysis

Table 1 reveals that both GPT-4o and LLAMA-3 exhibit hallucinations on almost all the examples. Regarding medical event inconsistency, GPT-4o had 21 summaries with incorrect information and 50 summaries with generalized information, while LLAMA-3 had 19 and 47, respectively. Both models produced many more "specific => general" errors, indicating the tendency of these models to produce less precise words that are potentially more frequently used in the general domain. We conjecture that domain-specific training will alleviate such errors. Among the 5 categories, most errors are from the symptoms/diagnosis and the medicinal instructions. This is unsurprising since these two categories involve the most expert knowledge and terminologies.

Considering "Incorrect Reasoning", LLAMA-3 hallucinates less compared to GPT-4o, with 26 summaries with related errors versus GPT-4o's 44. This is, however, worth noting that LLAMA-3 generates shorter summaries given the same prompt and does not attempt to infer additional details as much as GPT-4o. GPT-4o had an average of 516.46 words per summary with 27 summaries exceeding 500 words, while LLAMA-3 had an average of 266.56 words with no summaries exceeding 500 words. This aligns with findings from [3] that longer summaries tend to entail more hallucinations. We add to the finding that these errors often stem from incorrect reasoning in addition to medical event inconsistencies.

We qualitatively analyze the hallucination patterns related to "Incorrect Reasoning" by randomly sampling several medical notes. For brevity, we present a few representative examples in Table 2. Our observations reveal that LLAMA-3 often omits entirely the details that would have been generated with hallucinations by GPT-4o. When prompted to include these details in the summary and allowed to summarize without a word limit, LLAMA-3 tends to state the details as-is, with somewhat acceptable interpretations of these details. In contrast, GPT-4o often makes bold, two-step reasoning statements that can lead to hallucinations.

Chronological inconsistencies are relatively rare in both models as GPT-4o had 2 summaries with chronological inconsistencies, whereas LLAMA-3 had only 1. This may be due to the nature of the medical notes studied, which typically span a single patient visit and lack a strong timeline aspect. Further study on a longitudinal dataset could provide more insights into chronological inconsistencies.

## 6  Automatic Hallucination Detection

Human expert annotation of hallucinations is both expensive and time-consuming. To potentially automate this task and scale up the analysis of hallucination patterns across more summarization models, we explore an extraction-based system and an LLM-based system for hallucination detection.

### 6.1  Extraction-based System

In order to scale the hallucination detection, our work employs the use of a commercial solution, HYPERCUBE, to detect medical event hallucinations. Through a combination of medical knowledge bases, symbolic reasoning, and NLP, HYPERCUBE is able to build a comprehensive representation of patient documents. Given a medical document, HYPERCUBE produces a set of events, each having an associated type and its own collection of properties. Since HYPERCUBE identifies explicit spans of text as evidence for properties which are subsequently consolidated into a coherent set of events, the system will overtly seek to address overlapping and potentially conflicting information. Additionally HYPERCUBE leverages a set of medical knowledge bases in order to link the extracted events and properties to concrete definitions with semantic relationships to other concepts. As a consequence, the set of events HYPERCUBE produces is often complete and precise with respect to its medical domain model.

We consider the patient journeys extracted by HYPERCUBE, one from the original patient record and another from the LLM summary, denoted $E_d$ and $E_s$. Set-membership comparison is employed to compare these sets of extracted events. The hierarchical and semantic nature of the event types and property values allows for partial matching of events with varying specificity. Our system

| | Number of Summaries with Each Hallucination Type | | | | Number of Detected Hallucinations of Each Type | | | |
|---|---|---|---|---|---|---|---|---|
| | GPT-4o | | Llama-3 | | GPT-4o | | Llama-3 | |
| | Incorrect | Spec. ⇒ Gen. | Incorrect | Spec. ⇒ Gen. | Incorrect | Spec. ⇒ Gen. | Incorrect | Spec. ⇒ Gen. |
| **Medical Event Inconsistency** | **21** | **50** | **19** | **47** | **29** | **298** | **26** | **245** |
| - Patient Information | 3 | 2 | 7 | 6 | 3 | 2 | 7 | 6 |
| - Patient History | 1 | 15 | 1 | 15 | 1 | 28 | 1 | 30 |
| - Symptoms/Diagnosis | 13 | 42 | 6 | 43 | 18 | 128 | 6 | 101 |
| - Medicinal Instructions | 4 | 38 | 4 | 40 | 4 | 124 | 6 | 93 |
| - Followup | 3 | 14 | 6 | 12 | 3 | 16 | 6 | 15 |
| **Incorrect Reasoning** | **44** | n/a | **26** | n/a | **114** | n/a | **53** | n/a |
| **Chronological Inconsistency** | **2** | n/a | **1** | n/a | **3** | n/a | **1** | n/a |

Table 1: Summary of hallucination types and detected instances.

| Medical Note | GPT-4o | Llama-3 | Llama-3 (no word limit) |
|---|---|---|---|
| Vitals - T 97.7 120/78 51 18 99% RA, PTBD output 100 cc | - Discharge: Stable vitals, no distress, clear lungs, soft abdomen, no tenderness, PTBD in place, draining well. | Omitted | Vital signs: Temperature 97.3°F, blood pressure 119/71 mmHg, pulse 63 bpm, respiratory rate 18 breaths/min, oxygen saturation 96% on room air |
| The patient remained afebrile with mild elevated WBC prior discharge. | Pertinent Results: - Blood tests showed elevated WBC, low Hgb, and Hct. | Omitted | The patient had a mildly elevated white blood cell count (WBC), which will be monitored as an outpatient. |
| VITALS: 98.5 187/94 54 18 99% RA | - Discharge: Stable vitals, clear lungs, 1+ pitting edema in BLE, alert and moving all extremities. | Omitted | Vital signs: Temperature 98.5°F, blood pressure 187/94 mmHg, pulse 54 bpm, respiratory rate 18 breaths/min, oxygen saturation 99% on room air |

Table 2: Comparison of medical notes and summaries.

maps events that are missing or over-specified in $E_s$ as compared to $E_d$ as "incorrect" hallucinations, since these correspond to events in the summary that do not present in the original medical note. Meanwhile, "specific => general" hallucinations correspond to events that are under-specified with respect to $E_d$.

By mapping the events and property types produced by Hypercube onto the aforementioned *hallucination categories* (Section 3.2), we automatically detect hallucinations in LLM-produced summaries. We analyze 15 randomly sampled GPT-4o summaries and show a few detection examples in Table 3. We observe that Hypercube tends to over-estimate the number of hallucinations due to limitations of the knowledge base, such as, being case-sensitive and disallowing paraphrases or acronyms. Nonetheless, we observe that Hypercube can detect hallucinations that are otherwise missed by human experts. These findings suggest that Hypercube can be a helpful tool for an initial hallucination detection step, which can then be integrated with human expert review to enhance overall detection accuracy.

## 6.2 LLM-based System

The LLM-based model leverages the advanced capabilities of large language models to detect hallucinations in medical summaries. As LLMs have demonstrated the ability to self-correct, they might also be capable of detecting hallucinations when appropriately prompted. We employ a straightforward prompt for hallucination detection: "Detect all the inconsistencies between an original document and its summary. The inconsistency could be any details in the summary that could potentially have a different interpretation from the document, or vice versa."

We use this prompt with GPT-4o, providing the original document and its summary as input. This method generates significantly fewer candidate hallucinations compared to Hypercube. Further,

most of the detections identified by GPT-4o are false positives, primarily inconsistencies in writing style, such as abbreviations, as shown in Table 4. Despite this, there are notable cases where the detected hallucinations require complex reasoning, involving the consolidation of multiple pieces of knowledge that would otherwise conflict with each other.

We conjecture that LLMs have potential to accelerate hallucination detection due to their ability to perform complicated reasoning and identify inconsistencies. However, the current results indicate that domain-specific training or better prompting are necessary to improve the precision and recall of LLM-based hallucination detection systems.

## 7 Conclusion and Future Work

We propose the task of faithfulness hallucination detection in medical record summaries generated by large language models (LLMs) such as GPT-4o and Llama-3. These hallucinations, where generated content deviates from the input, pose significant risks in leading to misdiagnoses and inappropriate treatments. We develop a framework for annotating faithfulness hallucinations, categorizing their types and implementing a web-based annotation interface. Our pilot annotation study confirms the presence of various hallucination types produced by both closed-source and open-source LLMs. We further establish two automated hallucination detection baselines and show their limitations.

Future work should focus on improving automatic detection systems to mitigate the expense of human annotations as well as alleviating faithfulness hallucinations in healthcare AI. Given our initial evaluation of two summarization LLMs, it is imperative to broaden this study to include a wider range of open-source and proprietary models on more diverse medical tasks. By addressing the issue of faithfulness hallucinations, we can move closer to developing trustworthy AI systems that enhance patient care.

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

## A  Annotation Process

The annotation process was structured as follows:

(1) **Training Phase**: Clinicians participated in a training session, where they practiced annotating a subset of notes with guidance.

(2) **Annotation Phase**: Clinicians independently annotated the remaining notes. Each annotation included the identified inconsistency, its category, and the corresponding evidence or explanation.

- Reading Medical Records: Clinicians began the annotation process by reading the medical record using a search tool that supports abstraction and extraction of medical events and allows semantic search over these events.
- Reading Summaries: They then read the corresponding summary from the annotation tool and followed the instructions in the guidelines to label the hallucinations.
- Verification: Clinicians could go back and forth between the medical record and the summary, using the semantic search tool to carefully verify suspicious information.
- Average Annotation Time: The average time taken for annotating each medical note was 91.5 minutes per annotator.
- Quality Control: A quality control (QC) member reviewed the annotations and made corrections directly on top of the annotator's work. QC calculated a quality score based on a random 5% sample per person.

(3) **Consensus Meetings**: We hold regular meetings to discuss challenging cases and ensure consistent annotations. Discrepancies were resolved through group discussions, refining the guidelines as required.

## B  Automatic Hallucination Detection Analysis

The tables below discuss a few samples on how hallucinations were detected by extraction-based model HYPERCUBE in table 3 and by a LLM-based model GPT-4o in table 4.

| Medical Note | GPT-4o Summary | Hypercube Detection | Human Verification |
|---|---|---|---|
| ADMISSION LABS:
08:07AM BLOOD T3-37* Free T4-0.7* | Admission Labs: Elevated TSH (98), low Free T4 (0.7), elevated CRP | Admission Labs: Elevated TSH (98), low Free T4 (0.7), elevated CRP | YES |
| Pertinent Results:
FLUID CULTURE (Final ___:
___ ALBICANS,
PRESUMPTIVE IDENTIFICATION.
HEAVY GROWTH.
ANAEROBIC CULTURE (Final):
NO ANAEROBES ISOLATED. | Pertinent Results:
- Blood tests showed elevated WBC, low Hgb, and Hct. Fluid cultures indicated Candida albicans. | Pertinent Results:
- Blood tests showed elevated WBC, low Hgb, and Hct. Fluid cultures indicated Candida albicans. | YES |
| Hypothyroidism:
TSH was initially found to be 98. She was continued on 150 mcg daily. IV levothyroxine was considered; however, patient was unwilling to undergo IV placement and HCP was not established at that time. TSH was 33 and patient was continued on PO regimen at 137 mcg daily, with endocrinology input. | - Hypothyroidism: The patient was non-compliant with levothyroxine. TSH was initially 98 and decreased to 33 with continued PO levothyroxine. Follow-up with endocrinology was arranged. | Hypothyroidism: The patient was non-compliant with levothyroxine. TSH was initially 98 and decreased to 33 with continued PO levothyroxine. | NO |
| History of Present Illness:
Ms. ___ is a ___ w/ stage IIIA squamous cell carcinoma of lung, s/p completion of concurrent ___ ___,
also w/ a ... | History of Present Illness:
- Ms. [Redacted] is a patient with stage IIIA squamous cell carcinoma of the lung, post completion of concurrent chemotherapy and radiation therapy. | History of Present Illness:
- Ms. [Redacted] is a patient with stage IIIA squamous cell carcinoma of the lung, post completion of concurrent chemotherapy and radiation therapy. | PARTIAL |
| Past Medical History:
- Hypertension (HTN) | History of Present Illness:
- The patient is an elderly male with a history of HBV/HCV cirrhosis complicated by portal hypertension and hepatocellular carcinoma (HCC) status post resection. | History of Present Illness:
- The patient is an elderly male with a history of HBV/HCV cirrhosis complicated by portal hypertension and hepatocellular carcinoma (HCC) status post resection. | YES |

**Table 3: Samples of hallucination detection by Hypercube.**

| Medical Note | GPT-4o Summary | GPT-4o Detection | Human Verification |
|---|---|---|---|
| Chief Complaint: | Chief Complaint:
- Bright red blood per rectum (BRBPR) | Chief Complaint:
- Original: The section is blank in the original document
- Summary: "Bright red blood per rectum (BRBPR)" is listed, while the original document doesn't specify. | YES |
| Chief Complaint:
dyspnea
reason for transfer: IP evaluation | Chief Complaint:
- Dyspnea
- Reason for transfer: Inpatient evaluation | Chief Complaint:
- Original: "CC: dyspnea reason for transfer: IP evaluation."
- Summary: "Dyspnea; Reason for transfer: Inpatient evaluation." | YES |
| History of Present IIlness:
yo *M* with CAD, PAD, RAS, CHF (EF 20%) initially admitted for dyspnea x 2 weeks. Pt described symptoms of orthopnea and PND.
On        a CT was placed by thoracic surgery.
1500 ccs total drained on        . | History of Present IIlness:
- The patient, a male with a history of CAD, PAD, RAS, and CHF (EF 20%), was initially admitted for dyspnea lasting two weeks. Symptoms included orthopnea and PND. A chest tube was placed, draining 1500 ccs. | History of Present IIlness:
- Original: Detailed symptoms include "orthopnea and PND," and specific timeframe "initially admitted for dyspnea x 2 weeks."
- Summary: Mentions orthopnea and PND but doesn't specify the "two weeks" timeframe. Also, it states "A chest tube was placed, draining 1500 ccs," while in the original, it says "1500 ccs total drained on        ." | NO |
| Physical Exam:
On admission:
T 98.3, BP 130/73, HR 108, RR 18, O2 94% RA
Gen-NAD
HEENT- OP clear, sclera anicteric
CV- Regular rate, no murmurs
Pulm- CTAB
Abd- +scar, ostomy in place, min TTP at LUQ, no guarding/rebound
Ext- wwp, no edema
Neuro- A&Ox3
Prior discharge:
VS: 98.8, 89, 106/91, 18, 99% RA
GEN: Pleasant with NAD, Dobhoff tube in place.
CV: RRR, no m/r/g
PULM: CTAB
ABD: Right flank with

catheter to gravity drainage. Site with dressing and c/d/i. LUQ ostomy patent.
EXTR: Warm, no c/c/e | Physical Exam:
- On admission: T 98.3, BP 130/73, HR 108, RR 18, O2 94% RA. General: NAD. HEENT: OP clear, sclera anicteric. CV: Regular rate, no murmurs. Pulm: CTAB. Abd: +scar, ostomy in place, min TTP at LUQ, no guarding/rebound. Ext: wwp, no edema. Neuro: A&Ox3.
- Prior discharge: VS: 98.8, 89, 106/91, 18, 99% RA. General: Pleasant with NAD, Dobhoff tube in place. CV: RRR, no m/r/g. Pulm: CTAB. Abd: Right flank with catheter to gravity drainage. LUQ ostomy patent. Ext: Warm, no c/c/e. | Physical Exam:
- Original: Detailed vitals and assessments for admission and prior discharge.
- Summary: Summarized correctly but nuanced clinical details from admission and prior discharge exams are missing (e.g., exact vitals for the prior discharge). | NO |

**Table 4: Samples of hallucination detection by GPT-4o.**