# OpenReview forum: "Faithfulness Hallucination Detection in Healthcare AI"
_KDD.org/2024/Workshop/AIDSH — KDD-AIDSH 2024 Oral_

### Official Review · Reviewer_apaz · 2024-06-17
**A practical and innovative framework to detect faithfulness hallucinations healthcare AI**

**Rating:** 9
**Confidence:** 4

**Review:**

This study focused on the critical issue of faithfulness hallucinations in medical record summaries generated by large language models (LLMs) such as GPT-4o and Llama-3. They developed a framework which was crafted with input from clinicians and implemented using a web-based annotation tool for detecting faithfulness hallucinations, categorizing them into five distinct types. What’s more, a pilot study involving 50 medical notes was conducted to confirm the presence of various hallucination types in summaries produced by both closedsource and open-source LLMs. This study showed excellent quality, clarity, originality and significance. However, the limitations of this study are also evident. Only two LLMs including GPT-4o and Llama-3 were used. Medical record summaries in English were detected but other languages such as Chinese and Arabic were not. Finally, the research on detecting faithfulness hallucinations in large sample medical records should be conducted to validate the robustness of the framework proposed in this study.

---

### Official Review · Reviewer_4AoU · 2024-06-19
**Clear research questions definition and pipeline, but limited amount of data**

**Rating:** 7
**Confidence:** 3

**Review:**

Pros:

1. The paper is clearly written and well-structured. The research question is well-focused and clearly defined.

2. The collaboration with clinicians to craft data annotation guidelines ensures that the framework is clinically relevant and practical.

3. The use of both extraction-based and LLM-based systems for automatic hallucination detection provides a thorough evaluation of different approaches.

Cons:

1. The sample size of pilot study is too small. It seems like 50 clinical notes is not generalisable.

2. The paper lacks a necessary discussion on how this work differentiates itself from or adds to existing research on faithfulness hallucination problems.

3. The paper is a good technical report, but the AC should consider whether it is a suitable fit as a research paper for this workshop.

---

### Decision · Program_Chairs · 2024-06-28

Accept (Oral)